# An open-source perceptual crossing device for investigating brain dynamics during human interaction

**Stephen Estelle** *, **Kenzo Uhlig** , **Leonardo Zapata-Fonseca** , **Sébastien Lerique** , **Brian Morrissey, Rai Sato, Tom Froese**

Embodied Cognitive Science Unit, Okinawa Institute of Science and Technology, Okinawa, Japan

* stephen.estelle@oist.jp

**Data Availability Statement:** All the data files are available from the following repository: https://osf.io/9ecxu/files/osfstorage.

## Abstract

The Perceptual Crossing Device (PCD) introduced in this report is an updated tool designed to facilitate the exploration of brain activity during human interaction with seamless real time integration with EEG equipment. It incorporates haptic and auditory feedback mechanisms, enabling interactions between two users within a virtual environment. Through a unique circular motion interface that enables intuitive virtual interactions, users can experience the presence of their counterpart via tactile or auditory cues. This paper highlights the key characteristics of the PCD, aiming to validate its efficacy in augmenting the understanding of human interactions. Furthermore, by offering an accessible and intuitive interface, the PCD stands to foster greater community engagement in the realm of embodied cognitive science and human interaction studies. Through this device, we anticipate a deeper comprehension of the complex neural dynamics underlying human interaction, thereby contributing a valuable resource to both the scientific community and the broader public.

## Introduction

Research on social cognition, which explores how people make sense of others, has traditionally employed individualistic paradigms where passive and detached stimuli are used to observe how people make sense of others. However, social situations involving cooperative tasks, and the corresponding feeling of being together, can only truly manifest in the context of real-time social interaction. Second-person neuroscience studies have demonstrated that even the brain activity is distinctive when comparing dyadic interactions with passive and social observations [1]. Accordingly, it has been argued that every experimental setup assessing social interaction should include at least two people dynamically interacting, a shared context in which participants can both feel engaged, and data collection and analysis at the dyadic level [1]. One paradigm that fulfills such requirements is the Perceptual Crossing Experiment (PCE). In the PCE, pairs of participants interact through a minimalistic interface based on a continuous sensorimotor feedback loop consisting of hand-controlled movements and tactile feedback patterns (See Fig 1). Therefore, the paradigm allows a shared and continuous

**Funding:** This work was supported by OIST Proof of Concept Program - Innovative Technology Research Project (R8_37). The corresponding funding was given to TF, and the funders had no role in study design, data collection and analysis, decision to publish, or preparation of the manuscript.

dynamic experience for users and the corresponding dyadic data collection for researchers. Moreover, the motor and tactile patterns allow participants to simultaneously interact in an embodied way, which is the basis for having an engaging experience [2].

This research paradigm has demonstrated its replicability not only in cohorts of neurotypical adults from diverse cultural backgrounds, as evidenced by studies such as [3–6], but also among adolescent populations [7] and adults diagnosed with high-functioning autism [8]. This characteristic of replicability stands in stark contrast to the challenges currently faced in studies on inter-brain synchrony (IBS) involving real-time dual electroencephalography (EEG) recordings, as highlighted by [9]. The introduction of the Perceptual Crossing Device (PCD) in this study establishes a novel framework aimed at supplying a transparent, publicly available device for researchers working with haptics interested in the social domain to examine sensorimotor dyadic interactions in real-time. The novelty of the device enhances user engagement through an augmented space that provides immediate feedback on user movements and incorporates extensive haptic feedback, while the framework allows for compatibility with hyperscanning equipment to record the brain activity of participants concurrently. These features are designed to facilitate a more comprehensive understanding of embodied interactions by integrating the dynamics of two interacting brains.

The PCE interface allows each participant to manipulate an avatar within a one-dimensional circular space. An overlap between avatars in the space triggers a binary tactile feedback (on-off). The circular space is not visible for participants and each participant can encounter additional distracting objects: (i) a motionless static object positioned randomly, and (ii) a mobile object that "shadows" the movement of the partner's avatar at a fixed constant offset (See Figs 2–4). The key aspect of the PCE is that the tactile feedback is the same regardless of

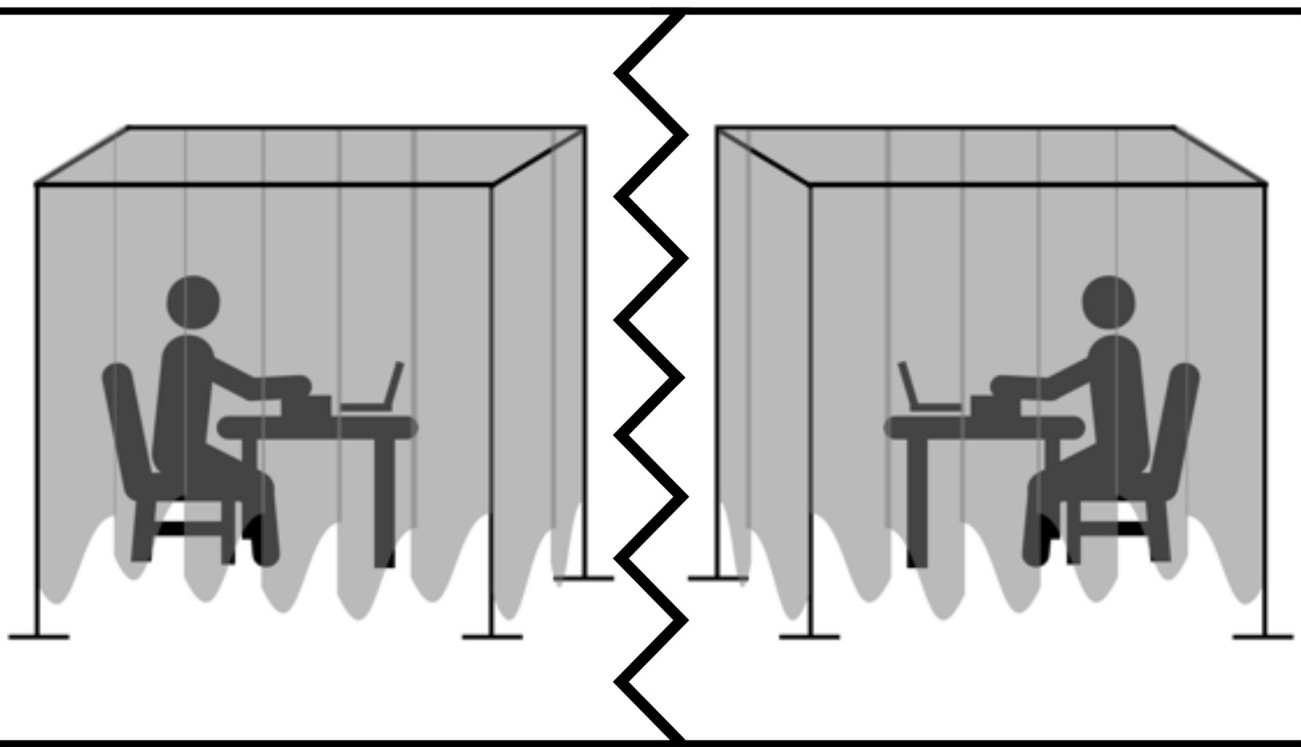

**Fig 1. Pair of participants in separate isolated locations operating the Perceptual Crossing Device.**

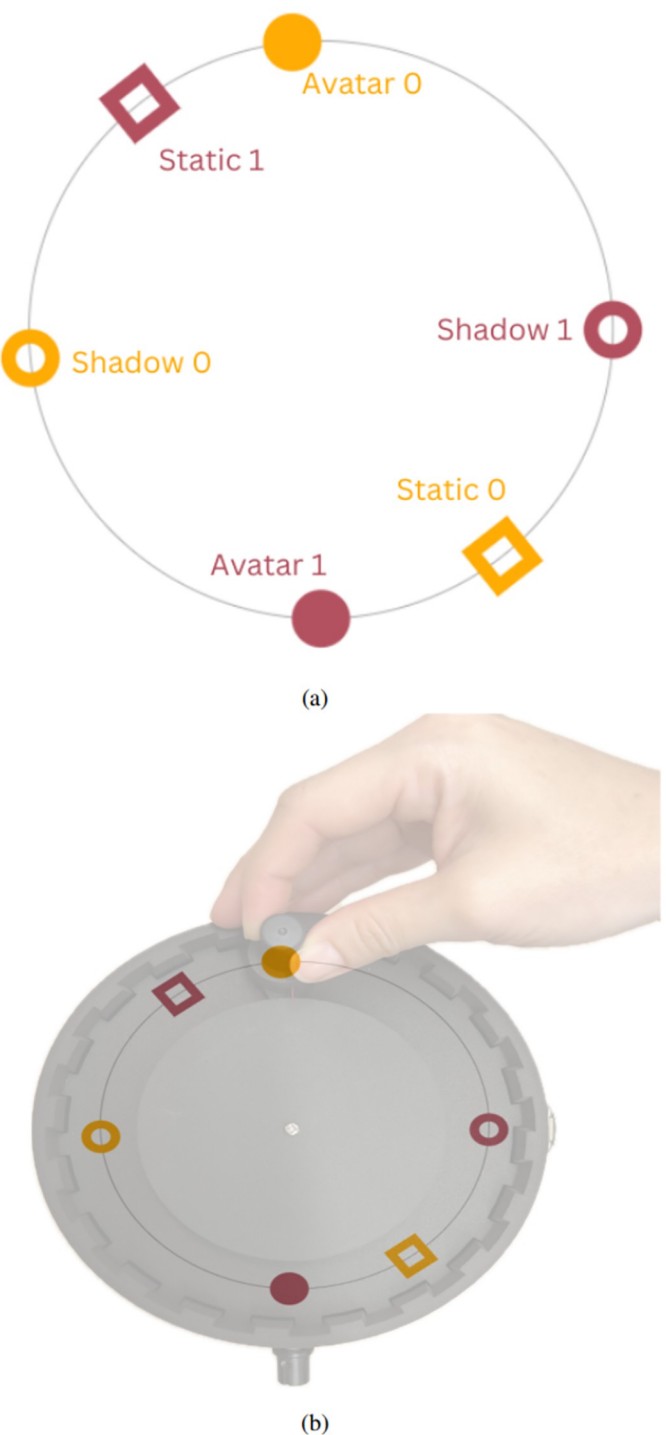

**Fig 2. Digital visualization of the perceptual crossing device's virtual space over the physical space.** (A) Virtual spacing. (B) Virtual space mapped to the controller.

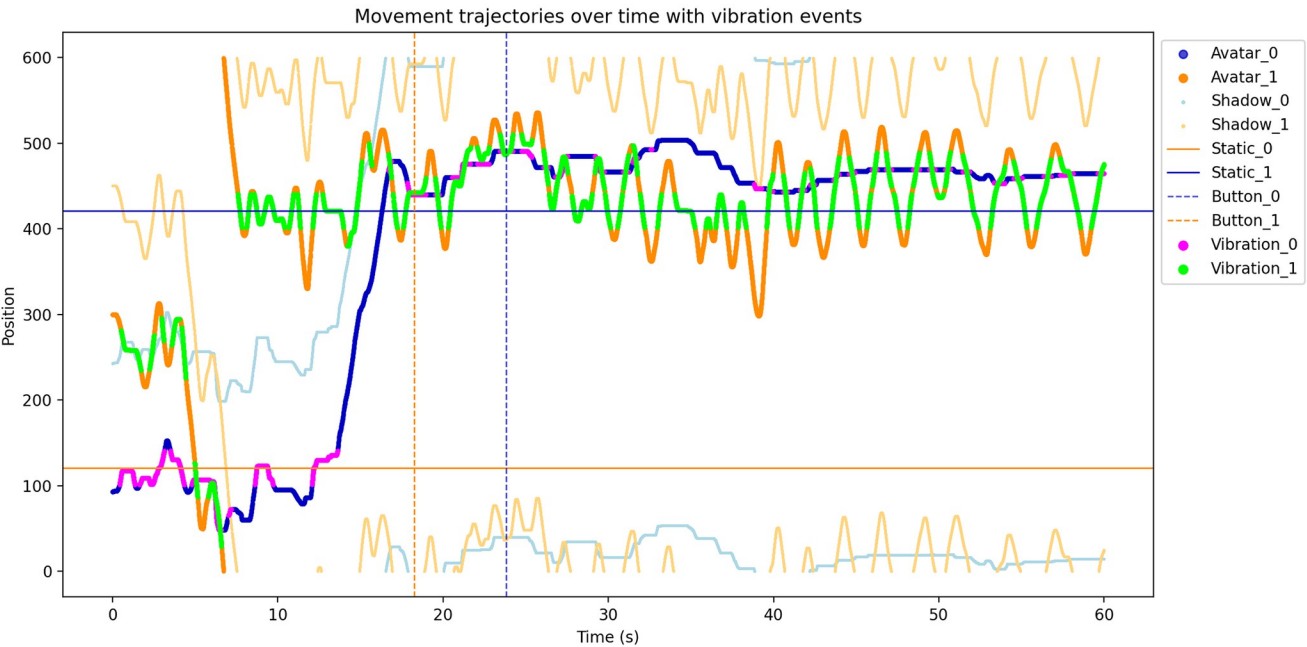

**Fig 3. Visualization of a recorded exemplary 60 s trial.**

| index | timestamp | static object 0 | static object 1 | motor 0 vibrate software | motor 1 vibrate software | pos0 | button0 | shadow delta0 | pos1 | button1 | shadow delta1 |
|---|---|---|---|---|---|---|---|---|---|---|---|
| 0 | 0.005562362 | 141 | 441 | 0 | 0 | 6.5625 | 0 | 150 | 300 | 0 | 150 |
| 1 | 0.001214152 | 141 | 441 | 0 | 0 | 6.5625 | 0 | 150 | 300 | 0 | 150 |
| 2 | 0.003075168 | 141 | 441 | 0 | 0 | 6.5625 | 0 | 150 | 300 | 0 | 150 |
| 3 | 0.005562362 | 141 | 441 | 0 | 0 | 6.5625 | 0 | 150 | 300 | 0 | 150 |
| . | . | . | . | . | . | . | . | . | . | . | . |
| . | . | . | . | . | . | . | . | . | . | . | . |
| . | . | . | . | . | . | . | . | . | . | . | . |
| 3100 | 6.559907816 | 141 | 441 | 0 | 1 | 120.9375 | 0 | 150 | 439 | 0 | 150 |
| 3101 | 6.562736839 | 141 | 441 | 0 | 1 | 121.40625 | 0 | 150 | 439 | 0 | 150 |
| 3102 | 6.570128514 | 141 | 441 | 1 | 1 | 121.875 | 0 | 150 | 439 | 0 | 150 |
| 3103 | 6.57148361 | 141 | 441 | 1 | 1 | 122.34375 | 0 | 150 | 439 | 0 | 150 |
| . | . | . | . | . | . | . | . | . | . | . | . |
| . | . | . | . | . | . | . | . | . | . | . | . |
| . | . | . | . | . | . | . | . | . | . | . | . |
| 29772 | 59.99300723 | 141 | 441 | 1 | 0 | 143.44 | 0 | 150 | 198 | 0 | 150 |
| 29773 | 59.99424275 | 141 | 441 | 1 | 0 | 143.44 | 0 | 150 | 197 | 0 | 150 |
| 29774 | 59.99756284 | 141 | 441 | 1 | 0 | 143.44 | 0 | 150 | 197 | 0 | 150 |
| 29775 | 59.99876968 | 141 | 441 | 1 | 0 | 143.44 | 0 | 150 | 197 | 0 | 150 |

**Fig 4. Perceptual crossing device collected data of a sample pilot trial.**

the object being overlapped. The usual task of this experiment is for participants to identify the moment of interaction with their partner's avatar and mark such an event with a button press. Crucially, the only way to distinguish between the objects is through patterns of interaction that participants deploy. When crossing the static and shadow objects, there is an unidirectional vibration that is only felt by the participant that is encountering those objects. In contrast, when both participants are crossing each other, the vibration becomes bidirectional, and both receive the feedback simultaneously.

**Table 1. Perceptual Crossing Devices.**

| PCD Version | Developer | Motor controller | Sensory feedback | Space mapping |
|---|---|---|---|---|
| Version 1 | Lenay 2006 [17] | Computer's mouse | Braille cells with pins | Arbitrary (left-right) |
| Version 2 | Froese et al. 2014 [5] | Trackball | One hand vibration | Arbitrary (left-right) |
| Version 3 | Current PCD 2024 | Knob on rotator | Two hands vibration | Circular 1:1 ratio |

As shown in Table 1, three main devices have been built since the first appearance of a PCD. In the table we only refer to the developers of the classical PCE versions, in which three objects in a one dimensional space are presented to each participant (as described previously and shown in Fig 2). Different studies with this set up have been carried out in various populations showing consistent results: participants can accurately distinguish the avatar from the other two objects, their performance increases over time and the behavior cannot be explained only at an individual level but rather the dyadic level of analysis needs to be considered ([3–8, 10]). In addition to this approach, several implementations have been used to further research on imitation, joint perception, strictly dyadic interaction, as well as on different sources of sensory feedback under these minimal conditions [11–14]. However, we consider that all of them have limitations. The biggest drawback of previous iterations is the lack of an immediate feedback for the user of their actual movements in the non-visible space [15]. The current PCD offers a circular shape and a knob on a rotating arm that allows the exploration of the space such that participant's movements are mapped 1-to-1 with the avatars' movements within the circular virtual space and the avatars move at the same speed as the spinning knob. This 1-to-1 immediate feedback is significant because it focuses on the social presence and social interaction between people, not the constitution of space and how they understand space. Additionally, the tactile feedback area in previous versions of the PCE was limited to the non-moving hand and to a rather small surface on the hand. The PCD presented here offers tactile feedback for both hands through the moving handle and the button-pressing handle. Employing both hands to enlarge the area of vibration enhances the stimulation, rendering it more engaging and providing a pronounced experience pertinent to social modalities. Furthermore, the implementation of the button-pressing handle increases the area of vibratory feedback considerably as it covers approximately two thirds of the palmar surface. In addition to tactile feedback, the current PCD offers audio feedback as well for use in situations where haptic feedback is not desired, giving options to the operator on how to perceive their interaction (See Fig 5).

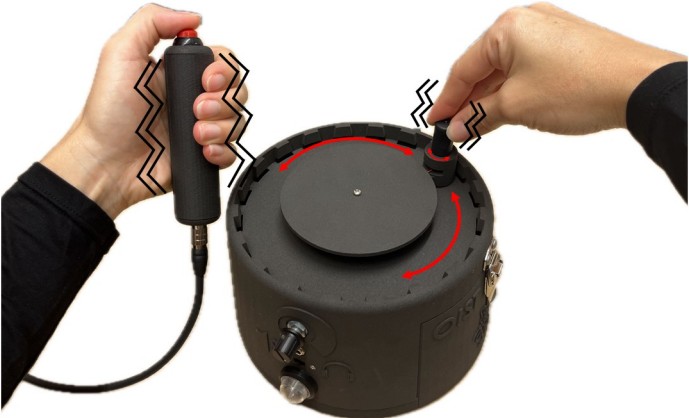

**Fig 5. Features of the Perceptual Crossing Device controller.**

The up-to-date model was designed to be easily distributed and replaceable for the community interested. This was made possible with 3D printing and circuit boards. Therefore, some, if not all the limitations posed by previous devices have been accounted for and overcame by this new PCD. Lastly, strict design specifications have been addressed, such as integration with a signal triggering system, due to the EEG and physio recording integration. Despite modifications to the PCD, previous findings could be replicated successfully, demonstrating that these changes did not adversely affect behavioral outcomes [16] (Preprint ECSU-PCE Dataset). Furthermore, unlike earlier versions of the PCD, details on validation and replicability have not been publicly disclosed until now. We are the first to provide such information, offering valuable resources for researchers, behavioral scientists, cognitive scientists, cognitive neuroscientists, and psychologists.

## The Perceptual Crossing Device system

The PCD system is engineered with a Raspberry Pi 4 Model B as its core processing unit, interfacing with two specialized PCD controllers (See Fig 6). This open-source system encompasses comprehensive documentation for assembly, including 3D models, a bill of materials, Printed Circuit Board (PCB) schematics, and requisite software scripts in Python and Elm languages. These resources are made available to facilitate replication and operation of the device by other researchers and practitioners.

Despite efforts made at facilitating the ease of replication of the PCD, it may present several technical challenges to those without an engineering skill set that may impact its widespread adoption, even with provided instructions. Key difficulties encountered in the assembly process include soldering, wire preparation, threading of 3D printed components, and software installation. However, the expectation is that the skill level of an undergraduate engineering intern would have the capability of assembling the entire PCD setup within 1 week.

Effective assembly of certain PCD components necessitates basic proficiency in soldering techniques. Additionally, the preparation of wires, involving precise cutting and stripping to specified lengths, is crucial for proper electrical connections. The 3D printed hardware components of the PCD require threading skills to ensure secure and consistent fastening. Finally, deploying the software onto the Raspberry Pi, an integral step for operational functionality, demands a foundational understanding of computer systems, in addition to the provided instructions.

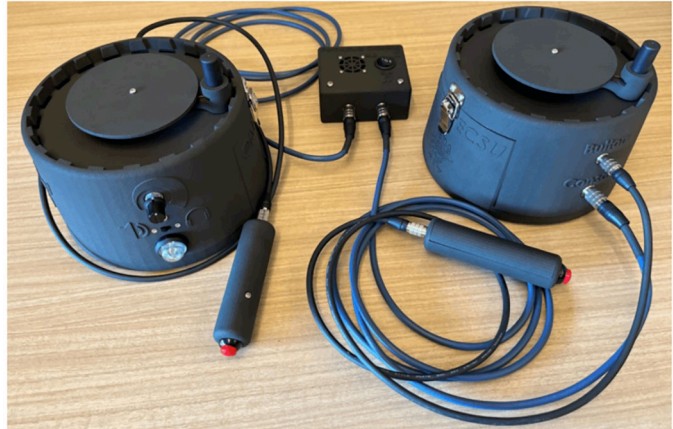

**Fig 6. Perceptual Crossing Device system (Console and 2 controllers).**

These challenges underline the need for specific technical skills and knowledge, potentially limiting the device's replicability among users with varying levels of expertise in electronics and computer science.

## Hardware

The PCD utilizes a rotary encoder (E6B2-CWZ3E-1024) for its circular motion, featuring 1024 pulses per rotation and a maximum permissible rotation speed of 6000 revolutions/minute. This encoder, with a 6 mm diameter shaft, is coupled to an A-type timing pulley (GPA32G-T2060-A-P7) and a 2GT timing belt (170-2GT-6) measuring 6 mm in width, with a pitch circumference of 170 mm and 85 teeth. This assembly is in turn connected to a centrally positioned B-type timing pulley (GPA40GT2060-B-P10) in the PCD controller. A 3D printed coupling component links the B-type pulley to a slip ring, which is a pivotal element for ensuring smooth, uninterrupted circular motion without wire tangling (See Fig 7). To stabilize the slip ring and the timing pulleys, the former is affixed to the controller's main 3D printed body, while the encoder is fastened to the base with an additional 3D printed component. The slip ring's motion is transmitted through a 3D printed handle, securely press-fitted onto its top shaft. The handle comprises three integral parts:

1. A polyamide handle (RG23SD-M4), 8 mm in diameter and 23 mm long, featuring a rubberized surface and a M4 x 7 mm insert screw for grip and stability.

2. A Linear Resonating Actuator (LRA) motor (235 Hz, 8 mm diameter), affixed to the slip ring, produces haptic feedback in the spinning handle. The LRA's rapid response to input signals is crucial for realistic tactile sensations with minimal delay.

3. A thrust needle roller bearing (BA0414) is situated at the handle's core, aiding in the assembly's smooth operation. This bearing, along with the top cover plate and other components, is secured using a M3 x 100 mm screw, creating a cohesive unit.

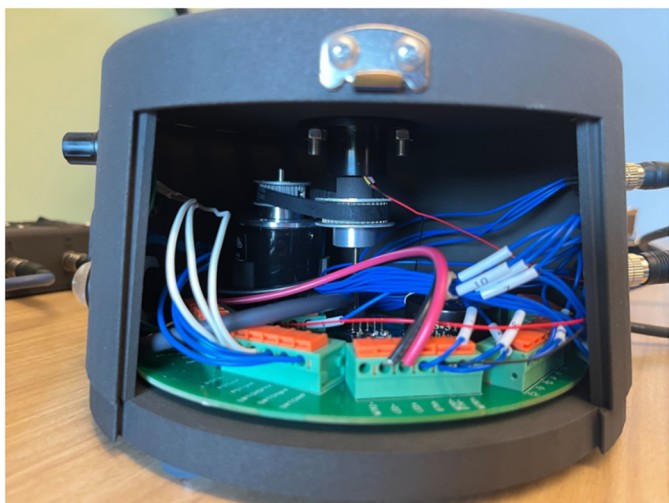

**Fig 7. Interior components of the Perceptual Crossing Device controller.**

The 3D printed parts, including the controller's main body, base, top cover plate, door, and handles, are fabricated from 3201PA-F Nylon using Selective Laser Sintering (SLS) technology, ensuring precise, lightweight, and durable components. However, the coupling part was 3D printed from Stereolithography (SLA) 3D printed resin for precision and extra durability. A panel-mounted LED (CNX718N20005W) on the controller's main body provides visual cues, complementing the haptic feedback. Audio capabilities are integrated into the controller, with components such as an aux jack (35RAPC4BV4), a speaker (CVS-3108), a potentiometer (RK0971110D88), and a 2-way switch (8SS1011-Z) for audio output management and mode selection between internal speakers and headphones.

The controller interfaces with the console and a button handle through 12-pin and 4-pin wires, respectively. The button handle houses an additional LRA motor and a button (P9-211121). Both LRA motors receive power from a single Adafruit Mono 2.5 W Class D Audio Amplifier (PAM8302).

The central console, housing the Raspberry Pi, orchestrates the PCD system. It includes a three-position rocker switch (COM-14978) to toggle between Haptic only, Haptic & Audio, and Audio only modes. An additional aux jack connects the console to a trigger box, facilitating integration with data collection systems.

Integration of trigger data with physiological measurements was achieved using the Brain Vision TriggerBox (Prod.-Rev.02). The acquisition of physiological data, encompassing modalities such as electroencephalogram (EEG), electrocardiogram (ECG), and respiration, was conducted using the BrainVision Recorder software (Version 1.20.0502) by Brain Products GmbH, Gilching, Germany. In this setup, the trigger signal from the PCD is captured concurrently with physiological data recording. This signal integration results in the generation of precise timestamps within the data stream. These timestamps are crucial for accurately synchronizing behavioral data obtained from the PCD with corresponding physiological data collected via the EEG recording system.

## Circuitry

The circuitry of the PCD is designed with an emphasis on simplicity, reproducibility, and ease of distribution. Within this framework, the LRAs are configured to generate vibrations following a square wave pattern at a frequency of 220 Hz. This specific frequency, as opposed to the resonant frequency of 235 Hz, is selected to minimize extraneous noise emissions when the motors are operational in conjunction with the Adafruit Audio Amplifier (See Fig 8).

The audio output system of the PCD is engineered to support a high-frequency response of 600 Hz and a low-frequency response of 300 Hz, whether through speakers or headphones. This dual-mode output provides users the flexibility to opt for noise-cancelling headphones for a more isolated auditory experience when necessary.

Incorporated within the console is a sophisticated trigger delivery mechanism. It is capable of outputting three distinct binary signals to a trigger box, thus enabling seamless integration with computer systems for data collection. These three outputs allow the PCD to transmit a combination of eight unique signals by alternating the binary state ("HIGH" or "LOW") of the pins in various configurations.

## Software

The software framework of the PCD System is a critical component that enables seamless interaction and data processing. A schematic of the system with its various components is shown in Fig 9. The Raspberry Pi in the center of the architecture runs the Python 3.9 scripts. The software leverages libraries such as pandas, scipy, gpizero, and pigpio for efficient data

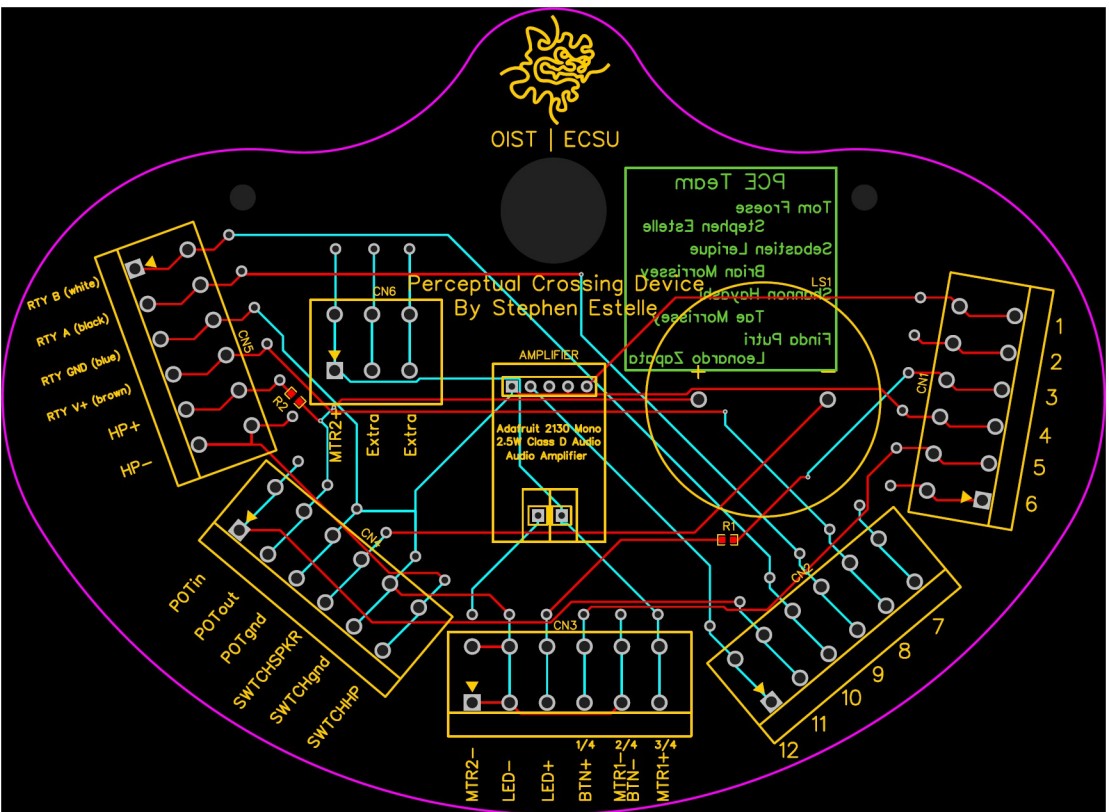

**Fig 8. Printed circuit board for the Perceptual Crossing Device that allows connectivity between all the interfaces.**

handling and GPIO (General Purpose Input/Output) control which is used for the communication with the two actual PCD. The system operates with a fixed refresh rate of 1 kHz, ensuring real-time responsiveness to user inputs. The actual consistent output frequency is subject of the latter validation section. The rotary encoder employed in the device utilizes 1024 ticks, and the software translates this information into a range of 0-599 in counter-clockwise direction, resulting in a total of 600 position units.

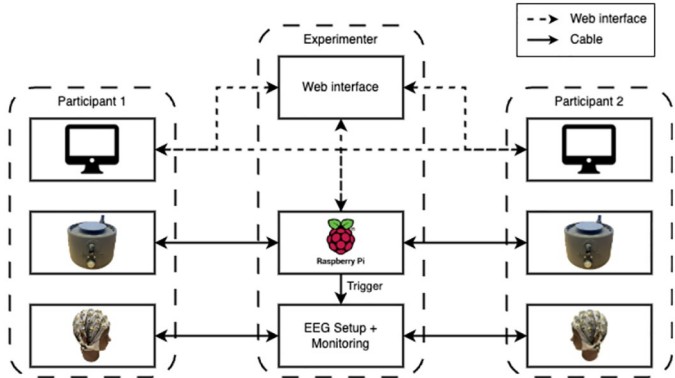

**Fig 9. Architecture of the Perceptual Crossing Device system for an exemplary experiment including two participants and an experimenter.**

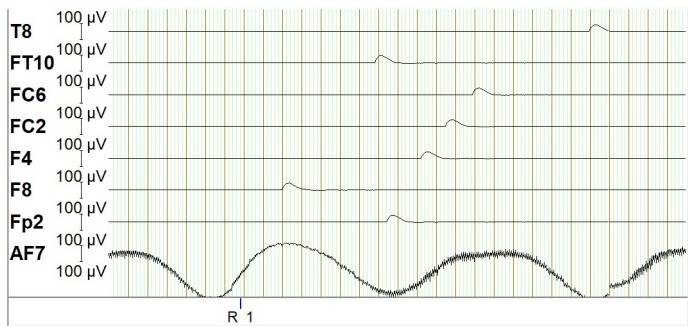

**Fig 10. EEG data along side a blue trigger from the PCD shown on the bottom of the graph.**

A distinctive feature of the PCD is its capability to visualize motion on a screen through a dedicated web interface as can be seen in the upper part of Fig 9. This interface is developed using Elm 0.19.1, providing a robust and user-friendly platform for interaction. The web interface serves as a valuable tool for experimental interaction. Experimenters can remotely communicate with participants in different locations, minimizing the need for the experimenter's physical presence. This functionality enhances the system's versatility in conducting experiments across various settings. Beyond experimental interaction, the web interface proves instrumental in participant training and monitoring. Experimenters can observe both the motion and button inputs in real-time. This feature not only facilitates participant training but also allows experimenters to remotely guide participants through experimental routines. The web interface becomes a central hub for overseeing and controlling experimental sessions, adding a layer of flexibility to the experimental setup.

To expand the capabilities of the PCD system, it can be seamlessly extended with EEG as shown in the bottom part of Fig 9. Specifically, the PCD delivers a trigger to the EEG software to indicate the start and end of trials and resting periods as indicated in Fig 10 to enable synchronization of the recorded data of the PCD and the EEG. This integration allows for the measurement of participants' brain activity, providing valuable insights into the cognitive processes underlying perceptual crossing interactions.

Fig 4 delineates the output data from the PCD. The data, formatted as a CSV file, encompasses variables such as position, time, vibration, and button status. Each data point, denoted as 'index', commences from 0 and represents a discrete observation from the PCD. The 'timestamp' variable records the time of data capture, ranging from 0 to 60 seconds within a single trial.

Variables 'static object 0' and 'static object 1' specify the locations of static objects corresponding to player 0 and player 1, respectively, within the circular arena. Motor vibrations are tracked via 'motor 0 vibrate software' and 'motor 1 vibrate software', employing binary values where 0 signifies absence and 1 indicates presence of a vibration signal from the software. Vibration signals are triggered when an avatar approaches within 20 units of another avatar, shadow, or static object.

The positions of the avatars are recorded as 'pos0' and 'pos1'. The relative positions of their shadows are indicated by 'shadow delta0' and 'shadow delta1', with a shadow delta value of 150 representing a shadow 150 units ahead of the avatar's position. Finally, 'button0' and 'button1' utilize binary values to reflect player button presses, where 0 indicates an unpressed state and 1 a pressed state, signaling the player's detection of another player.

The software framework of the PCD is designed to provide a robust, real-time interaction platform, enhancing its applicability in both experimental and training scenarios. To achieve this, initial software setup is required, along with the relevant supplies shown below:

- Raspberry Pi B

- Micro HDMI to HDMI Cable

- USB Type-C Power Cable

- 32GB (Class 10, U1) microSD card

- microSD card USB Reader

- USB Keyboard (for Raspberry Pi B)

- USB mouse (for Raspberry Pi B)

- Monitor (for Raspberry Pi B)

- Computer

- Internet Access (WiFi or Ethernet)

Maintenance requirements for the PCD are minimal. The primary maintenance activity involves ensuring that the screws securing the centrally positioned B-type timing pulley (GPA40GT2060-B-P10) and the 3D printed coupling component, which links the B-type pulley to the slip ring, are adequately tightened. Furthermore, it is crucial to ensure that the LRA motor within the spinning handle is securely affixed to the inside of the handle to allow proper haptic feedback. Full instructions encompassing software setup, hardware assembly, circuit board fabrication, and production of 3D printed components, is provided. Additionally, necessary 3D printing files along with their respective blueprints are available too. All of these documents are given on our repository:(https://osf.io/9ecxu/files/osfstorage).

## Performance validation

In this section, we present a systematic approach to validating the PCD, focusing on key components of its functionality. Fig 11 shows a schematic representation of this validation process which encompasses an examination of both input and output components, crucial for ensuring the reliability and accuracy of the device. Our validation process assesses the latency of these components within the system, vital for dictating user perception and motor coordination when interacting with the PCD. First, we validate the input into the system, specifically speaking, the precise recording of button presses. Then, the validation continues with the device's output, wherein we evaluate the audio and vibration feedback. Lastly, we investigate the resulting sampling rate of the Data Log, providing transparency regarding the expected data quality when utilizing the device.

To ensure the accurate integration of various components of the PCD, the Brain Products STIMTRAK Acoustical Stimulator Adapter was employed, chosen for its noted high precision in event marker generation. A marker acts as a timestamp that shows the exact time when events occurred in line with the EEG data. Empirical measurements indicate a marginal deviation of approximately 0.25 ms between the StimTrak-generated markers and the simulation markers [18].

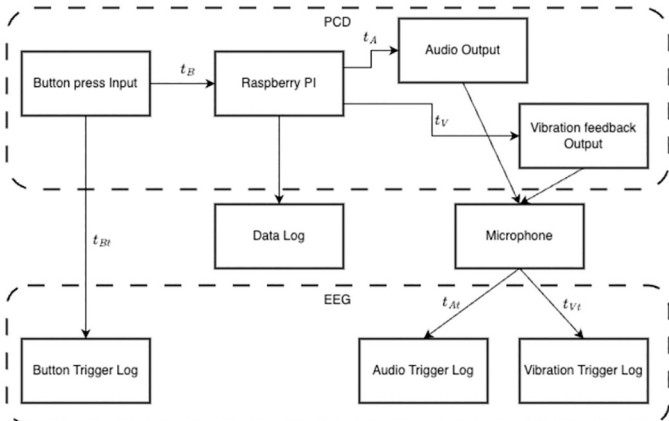

**Fig 11. Design of validation of the PCD.**

## Button latency

The latency $\Delta t_{Bl}$ between the actual actuation of a button at time $t_{Bt}$ and its corresponding recorded timestamp $t_B$ in the data-log can be calculated with $\Delta t_{Bl} = t_B - t_{Bt}$.

To capture the real-time button press, the Brain Products STIMTRAK Acoustical Stimulator Adapter was utilized. This adapter facilitated the transmission of an output signal to the trigger box, which was interfaced with the EEG recording system's trigger box. Upon pressing the PCD controller button, a change in voltage occurs, triggering a signal to the EEG system. This event's recorded timestamp was then cross-referenced with the button press data logged by the Raspberry Pi.

After analyzing 704 samples of button press, we observed an average button press latency of 1.9 ms, a median time latency of 1.9 ms and a calculated standard deviation of 9.3e-4. The corresponding histogram is shown in Fig 12. The magnitude of the latency is comparable to the

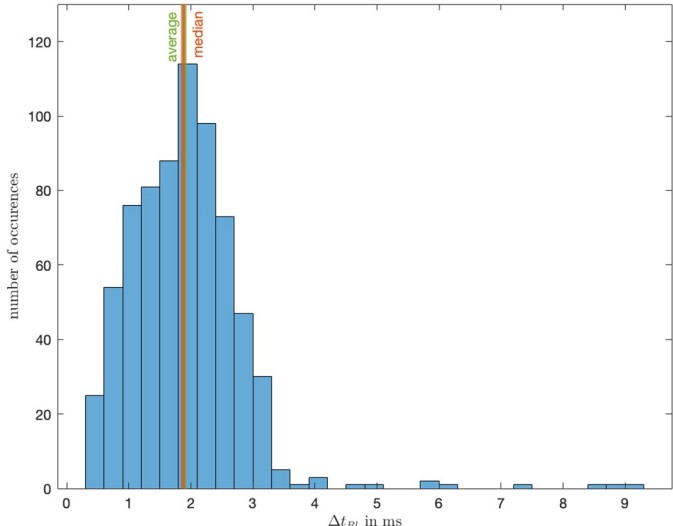

**Fig 12. Latency between button press and timestamp for the PCD controller.**

peripheral click latency of commercially available computer gaming mice which is around 2 ms [19].

## Tactile and audio stimulation

**Tactile stimulation.** The validation of tactile stimulation in the PCD focused on quantifying the latency $\Delta t_{Vl}$ between the initiation of a trigger signal for motor vibration $t_V$ and the actual onset of tactile stimulation $t_{Vt}$ experienced by the user, which were determined using the microphone integrated into the Brain Products STIMTRAK Acoustical Stimulator Adapter. The microphone was tasked with detecting the commencement of motor vibration, generating a corresponding trigger signal that was then synchronized with the recorded data timestamps. For precise measurement, the device was positioned adjacent to the motor on the spinning handle, and its sensitivity was adjusted to accurately capture the motor's activation moment.

The latency of the tactile stimulation can then be calculated using $\Delta t_{Vl} = t_{Vt} - t_V$.

An analysis of 112 samples revealed that the motor exhibited an average latency of 16.2 ms, with a median latency of 14.1 ms. The standard deviation observed in these measurements was 13e-3. The corresponding histogram is shown in Fig 13.

**Audio stimulation.** In a procedure identical to the validation of tactile stimulation, our objective was to quantify the latency $\Delta t_{Al}$ between the generation of the audio output signal $t_A$, as recorded in the output data, and the time $t_{At}$ at which the sound becomes audible. This measurement was facilitated by employing the microphone in the Brain Products STIMTRAK Acoustical Stimulator Adapter, which is configured to send a trigger to the EEG data acquisition software upon detecting sound from the controller. The acoustical adapter was strategically placed inside the PCD controller near the speaker, with its sensitivity finely tuned to exclude extraneous noise. Analogously, with $\Delta t_{Al} = t_{At} - t_A$ the latency can be calculated.

Upon analyzing a sample of 862 instances, we determined that the average latency from the initiation of the audio signal to the occurrence of actual auditory stimulation was approximately 3.2 ms, with a median valued at 3.2 ms. The observed standard deviation was determined as 9.93e-4. Fig 14 shows the corresponding histogram.

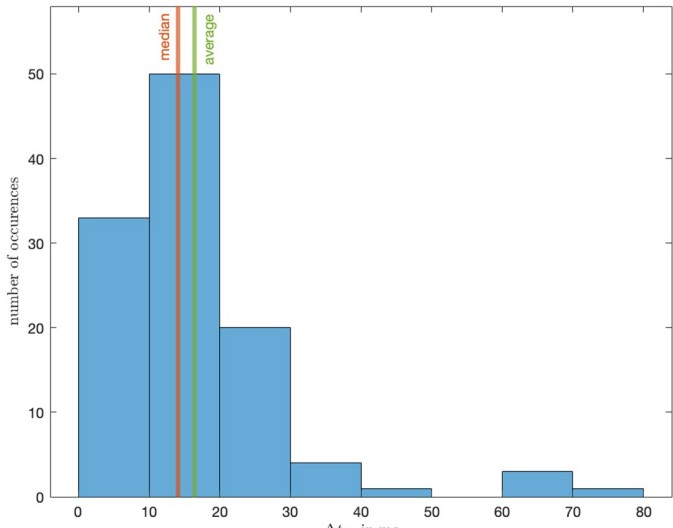

**Fig 13. Latency between tactile feedback and timestamp for the PCD controller.**

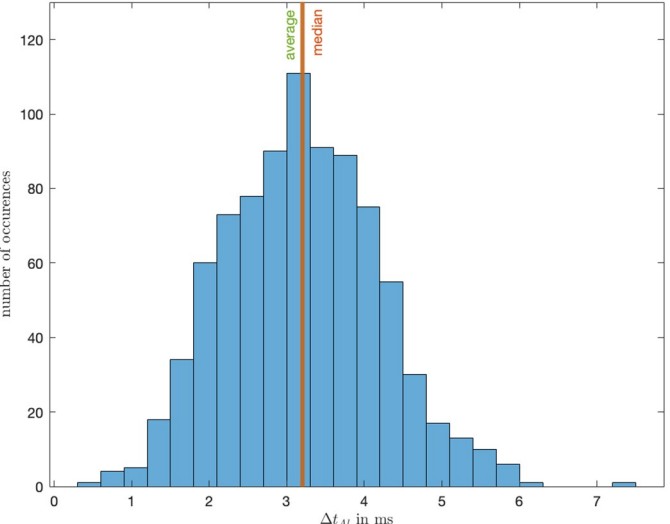

**Fig 14. Latency between audio feedback and timestamp for the PCD controller.**

The magnitude of both tactile and audio feedback latency are very quick and satisfy our technical expectation. By showing that the signal outputs are in the range we can be confident that the participant receive stimuli instantaneously. Compared to the tactile feedback the audio feedback is quicker which was expected due to the small but higher inertia of the motor which needs to ramp up to produce the vibration.

## Encoder resolution

The movement of the avatars within the system is controlled through a rotary encoder. This encoder governs the relative positioning of the avatars, as opposed to tracking their absolute spatial coordinates. A limitation to consider is the encoder's resolution, which is specified at 1024 pulse/rotation. Given that a complete revolution in the virtual environment equates to 600 units, it can be deduced that each pulse from the encoder corresponds to a movement of approximately 0.588 units (0.35˚) in the virtual space. Specifically regarding the operational precision of our device, the circumference of the spinning handle's field of movement on the controller measures 400.55 mm. With the encoder configured to 1024 pulses/rotation, each pulse translates to approximately 0.39 mm along the circumference, which constitutes roughly 0.1% of the total circumference. Given this precision, we assessed that, although more accurate encoders are available, the resolution provided by our current encoder is sufficiently precise for detecting human movements in the experimental setup.

## Sampling time

The analysis of the sampling time aimed to ascertain the resolution of data logging under conditions simulating realistic usage and full computational load on the Raspberry Pi. The Raspberry Pi's sampling frequency is configured to 1 kHz, setting an upper limit for expected values. Nonetheless, considering the Raspberry Pi's role as the primary computation unit, a sampling rate lower than 1 kHz was anticipated.

An analysis of 534,168 data points yielded an average time increment of 2.02 ms and a median of 1.81 ms, all corresponding to a sampling rate of approximately 494.6 Hz as shown

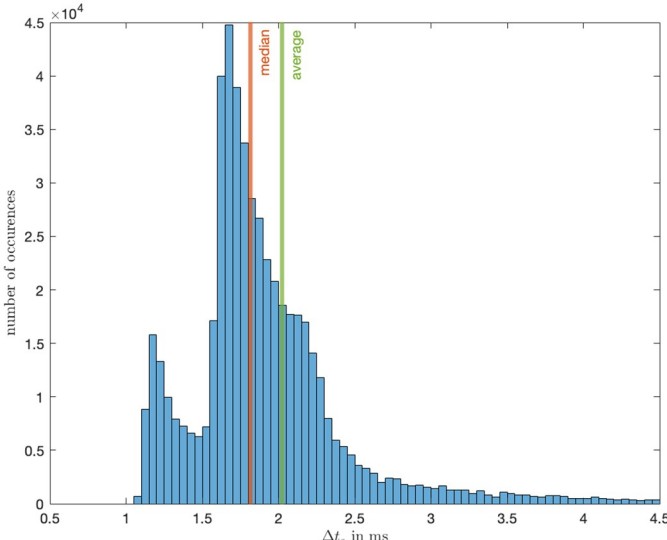

**Fig 15. Sampling frequency for the PCD of the data it collects.**

in Fig 15. The standard deviation in the time increments was recorded to be 1e-3. The resulting scale of sampling time is more than sufficient for the device's purpose to record human hand movements.

## Overall

The conducted validation tests on the PCD have affirmed its accuracy and efficacy. These tests demonstrate that the device can provide near real-time feedback, with the haptic motors and audio stimuli exhibiting average delays of 16.2 ms and 3.2 ms, respectively. This temporal precision is critical for researchers and practitioners conducting trials using the PCD and shows the real-time feedback to be shorter than the average latency detectable by humans for haptic and audio feedback at 205 ms and 245 ms respectively [20].

Furthermore, the device's button-press mechanism effectively captures the moment of perceived presence between users, evidenced by an average delay of only 1.9 ms (See Fig 16). Despite the encoder's limitation to a resolution of 1024 pulse/rotation (equivalent to 1 pulse per 0.35 degrees), the high sampling rate of approximately 494.6 Hz ensures accurate tracking of an individual's position throughout the experiment (See Fig 4).

These validation results underscore the PCD's potential as a reliable and replicable tool for use in hyperscanning research, providing researchers with precise and timely data essential for their studies.

## Future work

The future development of the PCD aims to enhance its accessibility as a replicable tool for EEG hyperscanning research. To achieve this, simplification of the PCD's hardware assembly process is essential. Potential improvements could include integrating pre-wired components or designing a console that facilitates straightforward assembly through clearly delineated plugs and jacks. Additionally, considering the outsourcing of manufacturing to professional entities could ensure the delivery of fully assembled devices to researchers, thereby mitigating the risks associated with user-assembled hardware.

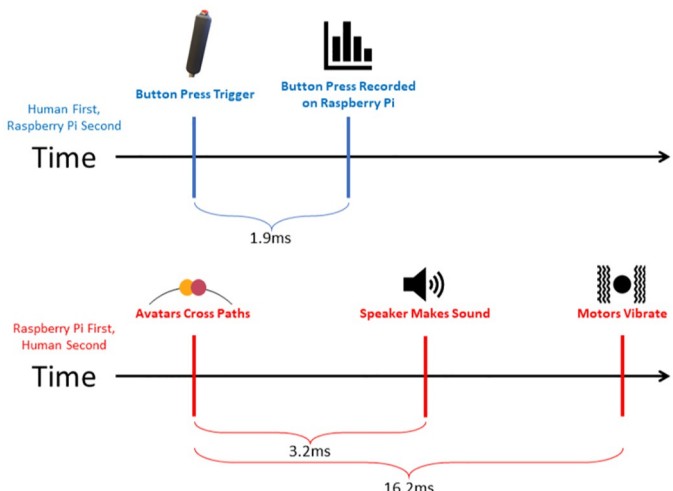

**Fig 16. Timeline of sequences in the performance validation.**

Advancements in software are equally crucial, particularly in developing a user interface (UI) that offers comprehensive customization capabilities for experimental trials and programming. Enabling researchers to tailor the PCD's functionality to their specific needs can foster innovation in experimental design and broaden the range of potential studies. An intuitive and customizable UI would significantly expand the user base of the PCD. In essence, optimizing the PCD for ease of acquisition and maximal customization aligns with the overarching objective of enhancing its utility and applicability within the scientific community.

Ultimately, the PCD leverages open-source software customization and replicable hardware design to foster innovation in experimental methodologies and data analysis within its field. This flexibility allows researchers to modify the code to suit their unique experimental needs, enabling the generation of innovative insights that are reproducible using the same open-source equipment. Moreover, the PCD builds on the legacy of prior devices, enhancing the scope of ongoing research discussions. An example includes extending the work of Lenay and Stewart (2012) [4], who utilized audio feedback in a perceptual crossing experiment, thus demonstrating the PCD's capacity to integrate and expand upon existing research frameworks.

The PCD offers potential applications beyond its initial scope as well. Notably, in the field of hyperscanning, where each study typically employs its own unique system for data collection, the lack of standardization complicates the replication and comparison of data across studies. The PCD could provide a standardized solution that enhances data uniformity, thus improving community-wide data sharing and validation within the hyperscanning field.

Furthermore, the PCD holds promise for advancing research into social disorders such as schizophrenia and autism [8, 21]. Although preliminary studies have explored the utility of similar technologies in these areas, further investigation using the PCD could significantly contribute to our understanding and support of these conditions.

## Acknowledgments

We are grateful for the help and support provided by Hinako Irei, Finda Putri, Tae Morrissey, Shannon Hayashi, and the Mechanical Engineering section of Core Facilities at Okinawa Institute of Science and Technology Graduate University.

## Author Contributions

**Conceptualization:** Stephen Estelle, Kenzo Uhlig, Leonardo Zapata-Fonseca, Sébastien Lerique, Tom Froese.

**Data curation:** Stephen Estelle, Kenzo Uhlig, Leonardo Zapata-Fonseca.

**Formal analysis:** Stephen Estelle, Kenzo Uhlig, Leonardo Zapata-Fonseca.

**Funding acquisition:** Sébastien Lerique.

**Investigation:** Stephen Estelle, Leonardo Zapata-Fonseca, Sébastien Lerique, Tom Froese.

**Methodology:** Stephen Estelle, Kenzo Uhlig, Leonardo Zapata-Fonseca, Tom Froese.

**Project administration:** Stephen Estelle, Sébastien Lerique.

**Resources:** Stephen Estelle, Sébastien Lerique, Brian Morrissey.

**Software:** Stephen Estelle, Kenzo Uhlig, Leonardo Zapata-Fonseca, Sébastien Lerique, Rai Sato.

**Supervision:** Stephen Estelle, Sébastien Lerique, Tom Froese.

**Validation:** Stephen Estelle, Kenzo Uhlig, Leonardo Zapata-Fonseca, Brian Morrissey, Rai Sato.

**Visualization:** Stephen Estelle, Kenzo Uhlig, Leonardo Zapata-Fonseca.

**Writing – original draft:** Stephen Estelle, Kenzo Uhlig, Leonardo Zapata-Fonseca, Brian Morrissey.

**Writing – review & editing:** Stephen Estelle, Kenzo Uhlig, Leonardo Zapata-Fonseca, Brian Morrissey, Tom Froese.

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
