## [Decision Letter · Decision Letter 0]

10 Apr 2024

PONE-D-24-06303An open-source perceptual crossing device for investigating brain dynamics during human interactionPLOS ONE

Dear Dr. Estelle,

Thank you for submitting your manuscript to PLOS ONE. After careful consideration, we feel that it has merit but does not fully meet PLOS ONE’s publication criteria as it currently stands. Therefore, we invite you to submit a revised version of the manuscript that addresses the points raised during the review process.

 One expert reviewer provides below helpful comments toward improving the submitted work. These mainly relate to sharpening the clarity of the manuscript with respect to its intented audience, for instance by considering theoretical implications and mitigations of technical complications.  In addition to the comments of this reviewer, I would like to raise two additional remarks myself:- In the motivation of the study, it is reported that previous devices were limited at the location and area of vibrostimulation (lines 53-55). I recommend making clearer the reasons as to why this can be considered a limitation. - Line 230, there is a reference to a 'marker', but it is unclear what marker is being referred to. 

We look forward to receiving your revised manuscript.

Kind regards,

Dimitris Voudouris

Academic Editor

PLOS ONE

4. Please include your tables as part of your main manuscript and remove the individual files. Please note that supplementary tables (should remain/ be uploaded) as separate "supporting information" files

Reviewers' comments:

Reviewer's Responses to Questions

**Comments to the Author**

1. Is the manuscript technically sound, and do the data support the conclusions?

Reviewer #1: Partly

2. Has the statistical analysis been performed appropriately and rigorously? 

Reviewer #1: No

3. Have the authors made all data underlying the findings in their manuscript fully available?

Reviewer #1: Yes

4. Is the manuscript presented in an intelligible fashion and written in standard English?

Reviewer #1: Yes

5. Review Comments to the Author

Reviewer #1: OVERALL

The authors present designs and validation for a perceptual crossing device that support investigations of real-time human social interaction. I see this version of the Perceptual Crossing Device as an excellent example of engineering innovation applied to address critical and difficult questions in the social sciences. With that said, I have a few major concerns about the manuscript and a handful of minor concerns.

MAJOR

• A defining and celebrated feature of PCE is the theory-experiment loop, in which simulations of perceptual crossing inform and are informed by experiments with humans. How do the authors think the PCD contributes to this rich and ongoing dialogue? Does the PCD create opportunities for novel experimental designs or analysis?

• The authors say “The biggest drawback of previous iterations is the lack of an immediate feedback for the user of their actual movements in the non-visible space” (lines 48-49). Could the authors motivate this claim more? I can see an antagonistic perspective, which would argue that not having information about location or movement makes the task more difficult and therefore success via mutual interaction more impressive or interesting. By providing participants with additional environmental information, the narrative describing successful interaction becomes more complicated to disentangle. I’d like to understand why the authors think this particular complication will benefit investigations of social interaction, beyond being an interesting engineering innovation.

• While PLOS ONE is largely a generalist journal, the intended audience for this manuscript is unclear to me. If the authors have a more engineering-oriented audience in mind, then I would expect additional explication of the PCE, its contribution to the study of social interaction, and more details about previous PCDs. If the idea audience is psychologists studying social interaction, then it seems necessary to me to report whether the modifications to this PCD (e.g. providing access to spatial information to participants) influence behavior in any way. In either case, I would like to see at least a brief discussion of how the modifications and additions to this version of PCD will contribute to the literature on PCE specifically and social interaction generally. If the authors are aiming for a multidisciplinary reach, then all of these items are important.

• The authors note several technical challenges for those without specific engineering skills. How accessible are, for example, online tutorials (e.g. YouTube) for acquiring the necessary skills to implement the PCD? This could be a damaging limitation since many researchers who might be interested in building their own PCD for human experiments likely have training in psychological science as opposed to engineering or computer science. I think this is another instance where thoughtfully considering the intended audience is important.

• The community of researchers applying the perceptual crossing paradigm is not especially large. What applications might this device have outside of perceptual crossing?

MINOR

Introduction

• Lines 6 – 8: expand on the difference made by engaging, real-time social interactions and why this difference is meaningful/pressing for studies of social behavior and social cognition.

• Lines 9 – 12: It seems worthwhile to briefly state the broad goal of the PCE at this point, in addition to the format of the experiment. I note that the authors describe the task on lines 30 – 31, but making the broad goal clear early (and later providing additional details, which the authors have done) will help readers not only understand the context of the PCE but also the contribution of the PCD to the field more broadly.

• Lines 19 – 20: The authors say the new PCD “provides a novel framework.” Is the framework simply the integration of physiological data with a sensorimotor interaction task? The nature of the framework and what about it is novel is unclear to me.

Perceptual Device System

• Please report how the (absence of) delay in sensory feedback compares to previous iterations of the PCD.

• The authors provide instructions for 3D printing materials through an external company but also include CAD files for 3D printing. Do these CAD files contain all the necessary information to 3D print all parts oneself?

• In several instances throughout the paper, what should be open quotation marks (‘) are actually end quotation marks (’). In LaTeX, an open quote is achieved with `.

Performance Validation

• Line 223: typo (“more specific the precise recording of button presses”)

• The authors note that the tactile stimulation has a mean latency of 16.2 ms and the audio stimulation has a mean latency of 3.2 ms. Given that PCE largely has a psychologically-oriented audience of researchers, it would be worthwhile to include a reference confirming that both of these latencies are shorter than a latency that is detectable by or influential to humans.

• In-text caption for Figure 13 presumably has a typo. I assume it should say auditory feedback?

• The authors note a key limitation is the rotary encoder’s resolution. It would be helpful to comment on the extent to which this resolution is detectable by humans or could possibly influence behavioral or psychophysiological data.

• Low resolution for figures 3 and 9. Could the authors provide higher resolution images?

• Could the authors discuss the longevity of the PCD? What is involved in device maintenance? To what extent and at what rate might the signal quality degrade over time?

6. PLOS authors have the option to publish the peer review history of their article (what does this mean?). If published, this will include your full peer review and any attached files.

Reviewer #1: **Yes: **Haily Merritt

---

## [Author Response · Author response to Decision Letter 0]

13 May 2024

I would like to update the financial disclosure to match the funding information section.

"This work was supported by OIST Proof of Concept Program - Innovative Technology Research Project (R8_37). The corresponding funding was given to TF, and the funders had no role in study design, data collection and analysis, decision to publish, or preparation of the manuscript."

All other comments have been responded to in the attached file "Response to Reviewers"

---

## [Decision Letter · Decision Letter 1]

28 May 2024

An open-source perceptual crossing device for investigating brain dynamics during human interaction

PONE-D-24-06303R1

Dear Dr. Estelle,

We’re pleased to inform you that your manuscript has been judged scientifically suitable for publication and will be formally accepted for publication once it meets all outstanding technical requirements.

Kind regards,

Dimitris Voudouris

Academic Editor

PLOS ONE

Additional Editor Comments (optional):

Reviewers' comments:

Reviewer's Responses to Questions

**Comments to the Author**

1. If the authors have adequately addressed your comments raised in a previous round of review and you feel that this manuscript is now acceptable for publication, you may indicate that here to bypass the “Comments to the Author” section, enter your conflict of interest statement in the “Confidential to Editor” section, and submit your "Accept" recommendation.

Reviewer #1: All comments have been addressed

2. Is the manuscript technically sound, and do the data support the conclusions?

Reviewer #1: Yes

3. Has the statistical analysis been performed appropriately and rigorously? 

Reviewer #1: N/A

4. Have the authors made all data underlying the findings in their manuscript fully available?

Reviewer #1: Yes

5. Is the manuscript presented in an intelligible fashion and written in standard English?

Reviewer #1: Yes

6. Review Comments to the Author

Reviewer #1: (No Response)

7. PLOS authors have the option to publish the peer review history of their article (what does this mean?). If published, this will include your full peer review and any attached files.

Reviewer #1: **Yes: **Haily Merritt

---

## [Editor Report · Acceptance letter]

30 May 2024

PONE-D-24-06303R1 

PLOS ONE

Dear Dr. Estelle, 

I'm pleased to inform you that your manuscript has been deemed suitable for publication in PLOS ONE. Congratulations! Your manuscript is now being handed over to our production team.

Kind regards, 

on behalf of

Dr. Dimitris Voudouris 

Academic Editor

PLOS ONE